# Effect of Medetomidine, Dexmedetomidine, and Their Reversal with Atipamezole on the Nociceptive Withdrawal Reflex in Beagles

**DOI:** 10.3390/ani10071240

**Published:** 2020-07-21

**Authors:** Joëlle Siegenthaler, Tekla Pleyers, Mathieu Raillard, Claudia Spadavecchia, Olivier Louis Levionnois

**Affiliations:** 1Section of Anaesthesiology and Pain Therapy, Department of Clinical Veterinary Sciences, Vetsuisse Faculty, University of Berne, 3012 Bern, Switzerland; joelle.sieg@gmail.com (J.S.); tekla.pleyers@vetsuisse.unibe.ch (T.P.); mathieu_raillard@yahoo.it (M.R.); claudia.spadavecchia@vetsuisse.unibe.ch (C.S.); 2University Veterinary Teaching Hospital, School of Veterinary Science, Faculty of Science, The University of Sydney, Sydney 2006, Australia

**Keywords:** antinociception, atipamezole, dexmedetomidine, dog, nociceptive withdrawal reflex, sedation

## Abstract

**Simple Summary:**

Medetomidine, an alpha-2 agonist routinely used to provide sedation and pain relief in dogs, is a mixture of dexmedetomidine and levomedetomidine in equal proportions. Dexmedetomidine, considered to be the only active component in the mixture, is also marketed alone. Sedation caused by both formulations can be reversed using atipamezole, which shortens recovery. Dexmedetomidine provides analgesic effects similar to medetomidine, but it remains unclear at which dose and whether the analgesic effects of medetomidine or dexmedetomidine disappear once atipamezole is injected. The present trial aimed at elucidating these uncertainties using the nociceptive withdrawal reflex model. This model allows for quantification of analgesia by measuring specific activity from muscles involved in limb withdrawal in response to mild electrical stimulation. In eight beagles, the model was applied to compare the extent of pain relief provided by medetomidine and dexmedetomidine and to investigate whether complete reversal occurs after the administration of atipamezole. No difference in analgesic efficacy was identified between the two formulations. Both sedation and pain relief terminated rapidly when atipamezole was administered. These findings indicate that medetomidine and dexmedetomidine provide comparable levels of pain relief and that additional analgesics may be necessary when atipamezole is administered to dogs experiencing pain.

**Abstract:**

The objectives were: (1) to compare the antinociceptive activity of dexmedetomidine and medetomidine, and (2) to investigate its modulation by atipamezole. This prospective, randomized, blinded experimental trial was carried out on eight beagles. During the first session, dogs received either medetomidine (MED) (0.02 mg kg^−1^ intravenously (IV)] or dexmedetomidine (DEX) [0.01 mg kg^−1^ IV), followed by either atipamezole (ATI) (0.1 mg kg^−1^) or an equivalent volume of saline (SAL) administered intramuscularly 45 min later. The opposite treatments were administered in a second session 10–14 days later. The nociceptive withdrawal reflex (NWR) threshold was determined using a continuous tracking approach. Sedation was scored (0 to 21) every 10 min. Both drugs (MED and DEX) increased the NWR thresholds significantly up to 5.0 (3.7–5.9) and 4.4 (3.9–4.8) times the baseline (*p* = 0.547), at seven (3–11) and six (4–9) minutes (*p* = 0.938), respectively. Sedation scores were not different between MED and DEX during the first 45 min (15 (12–17), *p* = 0.67). Atipamezole antagonized sedation within 25 (15–25) minutes (*p* = 0.008) and antinociception within five (3–6) minutes (*p* = 0.008). Following atipamezole, additional analgesics may be needed to maintain pain relief.

## 1. Introduction

Medetomidine and dexmedetomidine are alpha-2 adrenoreceptor agonists routinely used to produce sedation and analgesia in dogs [1,2,3,4]. Medetomidine is a racemic mixture of two optical isomers, levomedetomidine and dexmedetomidine [5]. As dexmedetomidine is considered the only active isomer, it is commonly administered at half of the administered dose of medetomidine to obtain similar levels of sedation [6,7] and antinociception [4,6]. However, contrasting evidence indicates that dexmedetomidine induces weaker sedation [8] and stronger antinociception [3] than medetomidine, thus raising some questions about the common assumption of equipotency.

So far, the antinociceptive effects of medetomidine and dexmedetomidine have been mainly evaluated using behavioral models based on thermal, electrical or mechanical stimulation. These models typically rely on the direct observation of nocifensive reactions as end-points [3,4,6,9,10,11,12,13,14,15]. When used to evaluate antinociception induced by alpha-2 adrenoreceptor agonists, the concomitant sedation and muscle relaxation elicited by these drugs can introduce significant bias [2]. In addition, the process of nociceptive threshold determination can generally be performed only at relatively wide time intervals (every 10–30 min) [3,4,9,10,11,13,15]. A more accurate comparison of the antinociceptive properties of these drugs and their temporal characteristics might be obtained with the nociceptive withdrawal reflex (NWR) model [16]. Based on non-invasive transcutaneous electrical stimulation of peripheral nerves and surface electromyographic (EMG) recordings, it allows a reliable determination of the nociceptive threshold through the neurophysiological characterization of the evoked response. The NWR model, validated in several animal species [16,17,18], has been applied to quantify the antinociceptive effects of analgesic and anesthetic drugs [19,20,21], including alpha-2 agonists [22,23,24]. The recent introduction of an automated threshold tracking methodology, allowing an almost continuous NWR threshold determination, further improved the model [25], making it ideal for highlighting subtle temporal differences between drugs, as it would be required to reliably compare similar drugs such as medetomidine and dexmedetomidine.

Atipamezole, an alpha-2 adrenoreceptor antagonist, is commonly administered to reverse medetomidine- or dexmedetomidine-induced sedation and to shorten recovery. Previous work suggests that atipamezole might contemporaneously reverse antinociception [4,26], which can be considered a rather undesired property in clinical settings. To date, specific investigations on potential differential effects of atipamezole on sedation versus antinociception are scarce and do not provide conclusive evidence. Again, the NWR model could provide interesting quantitative data to fill this clinically relevant knowledge gap.

The aims of this study were: (1) to compare the effects of intravenous (IV) dexmedetomidine and medetomidine on sedation and antinociception in dogs, and (2) to investigate their differential modulation by intramuscular (IM) atipamezole. We intended to test the hypotheses that (1) dexmedetomidine at half the dose of medetomidine would evoke higher NWR thresholds than medetomidine using the continuous NWR threshold tracking method, and (2) following atipamezole administration, the NWR thresholds would return to baseline later than sedation scores.

## 2. Materials and Methods

This experiment was approved by the Cantonal Committee for Animal Experimentation (approval number 30356). During the study, the investigators were unaware of the treatments administered (blinded design).

### 2.1. Sample Size, Design

The first objective of the study was to detect a difference in peak NWR thresholds after administration of medetomidine or dexmedetomidine. In previous studies it was shown that the median NWR threshold of non-medicated dogs is approximately 2.5 (2–3) mA [19]. Medetomidine was expected to increase it by 5 (4–6) times (personal experience). A 20% difference between the NWR thresholds induced by dexmedetomidine compared to medetomidine was arbitrarily considered a relevant endpoint. With a crossover design, eight dogs at least were considered necessary (Wilcoxon signed-rank test, paired, two-tailed, effect size = 1.4, α = 0.05, 1-β = 0.9, GPower 3.1, Germany).

The second objective of the study was to detect a difference in duration of effect between antinociception and sedation after administration of atipamezole. Reversal of sedation is expected to occur within 10 (8–12) minutes [4]. A 10 min delay for reversal of antinociception was considered a relevant endpoint. With a crossover design, three dogs at least were considered necessary (Wilcoxon signed-rank test, paired, two-tailed, effect size = 5, α = 0.05, 1-β = 0.9, GPower 3.1, Germany).

A parallel study evaluating selected respiratory effects of medetomidine and dexmedetomidine using electrical impedance tomography (EIT) was also carried out at the same time (data reported elsewhere [27]). Absence of interference between EIT and NWR recordings was confirmed by manufacturers of both devices and during the experiment.

### 2.2. Animals

A total of eight intact experimental beagles (two females and six males) were enrolled in this study. Dogs were housed with other kennel-mates in groups of three, had access to an enriched indoor and outdoor course. They were not involved in any other experiments, and did not receive any drug other than anthelminthic in the two months preceding the trial. They were considered healthy based on physical examination and selected hematology and blood chemistry analysis. Total body weight (BW) was measured for each dog before each session. All the dogs had a body condition score of 5 or 6 on a scale of 9 (body condition system, Nestlé Purina, Vevey, Switzerland). Mandibular lymph nodes were moderately increased in size in five animals; this was attributed to mild dental disease without any sign of pain.

### 2.3. Randomization, Blinding and Treatment

Dogs were sedated twice (crossover design) with 10 to 14 days wash-out between the two sessions. Each session included two consecutive phases: (1) sedation at time zero (T_0_), and (2) reversal, 45 min later (T_45_). The sedation (Phase 1, T_0_–T_45_) consisted of either medetomidine (“MED”; Domitor, 1 mg mL^−1^; Orion Pharma, Switzerland; 0.02 mg kg^−1^ BW IV) or dexmedetomidine (“DEX”; Dexdomitor, 0.5 mg mL^−1^; Orion Pharma, Switzerland; 0.01 mg kg^−1^ BW IV). The reversal (Phase 2, T_45_–T_210_) consisted of either atipamezole (“ATI”; Antisedan, 5 mg mL^−1^; Orion Pharma, Switzerland; 0.1 mg kg^−1^ BW IM) or saline (“SAL”; NaCl 0.9%, B Braun Melsungen AG, Melsungen, Germany; volume equivalent to ATI, IM).

Before the experiment, dogs were randomly assigned (http://www.randomization.com) to one of four blocks (MED/ATI, MED/SAL, DEX/ATI, DEX/SAL, n = 2 per block) for the first session. The opposite treatments were administered during the second session. For example, a dog receiving MED/ATI at the first session would receive DEX/SAL at the second. Treatments were prepared each day by a person not involved in the data collection but familiar with the study design; similar volumes of colorless solutions were administered so investigators were unaware of the treatments administered.

Experiments took place over a 4-week period in November and December, 2018. Each day, only one dog was sedated. A quiet, dedicated room within the same facility where dogs were housed was used. The dogs were acclimatized to the room and to the investigators the day before the experiment. A standard meal was provided in the night before; water remained available overnight. Experiments started around 09:00 a.m. for all dogs.

### 2.4. Animal Preparation

The day before the experiment, hair was clipped over stimulation and recording sites on the left hindlimb, at the site of insertion of IV cannulas and around the thorax for the EIT belt.

On the day of the experiment, dogs were weighed, the identification number (electronic chip) was checked and a physical examination was performed. If the animal presented with abnormalities at physical examination or developed unexpected complications during the study, it would be excluded and replaced. A eutectic mixture of local anesthetics (Emla cream 5%; Aspen Pharma GmbH, Switzerland) was applied over both right and left cephalic veins and covered with an occlusive dressing approximately one hour before placement of intravenous catheters (22 gauge, 25mm, Optiva 2IV Catheter; Smiths Medical International Ltd., Lower Pemberton, UK).

Electrode positioning was standardized to avoid differences among body region and biological function [28,29,30]. The stimulation and recording sites were shaved and degreased and gentle abrasion was performed with abrasive tape (Red Dot Trace Prep; 3M, Switzerland). Two self-adhesive stimulation electrodes (Bluesensor N; Ambu, Germany) were placed 0.5 cm apart over the left lateral plantar digital nerve with the anode in the distal position. Two self-adhesive recording electrodes (Bluesensor N; Ambu, Germany) were placed 0.5 cm apart over the left tibialis cranialis muscle, 3 cm distal to the knee joint. A self-adhesive ground electrode (Bluesensor VL; Ambu, Germany) was placed over the latero-distal femoral epicondyle. The electrodes were fixed with bandages to avoid displacement. Electrode impedance was measured continuously, and the electrodes were replaced if above 3 kΩ. The dogs were positioned and gently maintained in right lateral recumbency on a soft pillow.

At the end of the experiment (latest T_210_), the electrodes were removed and a layer of multipurpose antiseptic cream (Bepanthen Onguent; Bayer AG, Switzerland) was applied to avoid skin irritation at the electrode sites. A heparin-based cream (Hirudoid cream; Medinova AG, Switzerland) was applied at the sites of venous catheterization to prevent phlebitis. The dogs were monitored for a further 4 h to prevent complications, discomfort or re-sedation, before being re-introduced to their normal environment.

### 2.5. NWR Threshold Determination

After instrumentation, electrical stimulation and EMG recording were initiated using a dedicated unit (Dolosys pain tracker; Dolosys GmbH, Germany). Each stimulation consisted of a train of five rectangular pulses (1 millisecond, 200 Hz). The EMG was recorded at 1 kHz over 500 milliseconds, starting 100 milliseconds before stimulation. The 100-millisecond time window preceding stimulation was used to evaluate the EMG background noise (“noise range”). The time window between 30 and 100 milliseconds after stimulation was used to evaluate the NWR (“NWR range”). The response was considered positive when the interval peak Z score was above 10, meaning that the peak EMG amplitude within the NWR range had to exceed the mean EMG amplitude within the noise range by at least 10 times its standard deviation. The first stimulation started at 1 mA. The NWR threshold was then evaluated following an up-and-down bracketing design with maximal cut-off intensity set at 50 mA. It was possible to stop stimulation at any time in case of evident discomfort or pain (vocalization or escape movement). Stimulations were repeated every 10 s (with 30% interval randomization) changing intensity by steps of 0.3 mA. When the intensity increased or decreased three consecutive times, the step became 0.5 mA until the intensity changed direction again. A measurement was automatically discarded when the EMG amplitude exceeded 10 µV within the noise range. In this case, stimulation was repeated at the same intensity. The NWR threshold was automatically estimated after each stimulation through a logistic regression of the last 12 valid stimuli [25].

A stable baseline NWR threshold measurement for at least 5 min was obtained before the first treatment (phase 1). Medetomidine or dexmedetomidine were injected over a minute in the right cephalic IV catheter. The end of the injection was defined as T_0_. The threshold was then continuously determined until it returned to baseline, or until the dogs became intolerant to the experimental settings (impatient to move, not staying quiet or lying), or until 210 min after T_0_ (T_210_). If the measurements were discontinued before T_210_, the threshold was arbitrarily given the value of its baseline in the remaining time period to avoid missing data for group comparison.

### 2.6. Sedation

The depth of sedation was scored at baseline and every 10 min until the end of the experiments using a previously described descriptive scale [31]. Seven items were evaluated. The scores attributed to each individual item were summed to obtain a total sedation score ranging from 0 (no sedation) to 21 (deepest possible sedation). According to the scoring system, sedation was defined as mild with a score of 4 to 6, moderate with a score of 6 to 12 and profound with a score of 13 to 21. The same two investigators, both unaware of the administered treatment, always scored the depth of sedation together (agreement for each item score).

### 2.7. Physiological Variables

Heart rate (HR, beats minute^−1^), respiratory rate (*f*_R,_ breaths minute^−1^) and temperature (T, °C) were measured (thorax auscultation and rectal thermometer) and recorded every 10 min. Thoracic impedance was continuously recorded by electrical impedance tomography (Swisstom BB2, Swisstom AG, Switzerland) [27]. Blood samples (2 mL) were obtained from the left IV catheter into EDTA collection tubes at 2, 4, 8, 16, 30, 60, 90, 120 and 180 min after T_0_ for future pharmacokinetic analysis (not performed yet).

### 2.8. Data Analysis

To ease comparison between animals and treatments, the mean baseline NWR threshold was calculated over the 5 min immediately preceding drug administration. The mean relative NWR threshold (absolute threshold divided by its baseline) [32] was then calculated for every minute after T_0_.

Data are presented as median (interquartile range). In accordance with the recommendations of the American Statistical Association [33,34], differences in median are presented together with the *p*-value for the respective statistical test (Sigmaplot v14.0; Systat software Inc., San Jose, CA, USA). Since *p*-values below 0.001 could not be determined, they are reported as *p* < 0.001.

For phase 1 (T_0_–T_45_), differences between treatments (paired data, MED versus DEX) for baseline NWR thresholds, time to relative NWR threshold peak and peak intensity were tested using a Wilcoxon signed-rank test. Differences between treatments for NWR thresholds and sedation scores were tested using a two-way ANOVA for repeated measures, accounting for effect of time and treatment within each subject. Post hoc pairwise analysis was performed with a Holm–Sidak test.

For phase 2 (T_45_–T_210_), differences between treatments (paired data, ATI versus SAL) for relative NWR thresholds at T_44_, time to relative NWR threshold ≤ 1.5, time to sedation score ≤ 3 and comparison of the two latter variables were tested using a Wilcoxon signed-rank test. Differences between treatments for NWR thresholds and sedation scores were tested using a two-way ANOVA for repeated measures, accounting for effect of time and treatment within each subject. Post hoc pairwise analysis was performed with a Dunnett’s test.

Additionally, differences between MED/SAL (n = 4) and DEX/SAL (n = 4) subgroups and between MED/ATI (n = 4) and DEX/ATI (n = 4) subgroups for time to relative NWR threshold ≤ 1.5 and time to sedation score ≤ 3 were evaluated. Two-way ANOVA for repeated measures and Wilcoxon signed-rank tests were used. Correlation between NWR thresholds and sedation scores was tested using Pearson product moment and linear regression.

## 3. Results

The dogs were seven (5–8) years old and weighed 12.9 (11.7–15.8) kg. One female presented signs of proestrus during the second session of the trial but was not excluded. One male dog vomited shortly after administration of the alpha-2 adrenergic agonist on both sessions (once with medetomidine, once with dexmedetomidine), without further complication. All dogs tolerated well the experimental setting in lateral recumbency. Electrical stimulations could be continued in all dogs until return to a relative NWR threshold of one (0.9–1.1) within the duration of the experiment, except in one dog on one occasion (group MED/SAL) when measurements were stopped at T_210_ while the relative NWR threshold was still > 1.5 and sedation score was > 5.

### 3.1. Phase 1 (T_0_–T_44_, MED Versus DEX)

#### 3.1.1. Sedation

Sedation scores > 13 (profound sedation) were observed in all dogs within a minute after the end of the IV injection of both MED and DEX. Peak sedation scores of 17 (16–17) were reached 10 min after T_0_ in all dogs. Sedation scores varied significantly over time (*p* < 0.001), but were not different between treatments (MED versus DEX, *p* = 0.67, Figure 1a).

#### 3.1.2. Nociceptive Withdrawal Reflexes

Baseline NWR thresholds were similar between groups: 1.1 (0.9–1.3) mA for MED and 1.2 (1.1–1.5) mA for DEX (*p* = 0.313). Relative NWR thresholds varied over time (*p* < 0.001) but not between treatments (MED versus DEX, *p* = 0.317). Peak relative NWR thresholds were reached at seven (3–11) minutes for MED and six (4–9) minutes for DEX (*p* = 0.938). Peaks of 4.4 (3.9–4.8) times the baseline and 5.0 (3.7–5.9) times the baseline were reached for MED and DEX, respectively (*p* = 0.547, Figure 2a). This corresponded to a difference of 14%, which was below the predefined relevant endpoint of 20%.

#### 3.1.3. Physiological Variables

After treatment administration, the HR decreased from 106 (100–118) beats minute^−1^ at T_0_ to 45 (34–60) beats minute^−1^ at T_44_ for MED (*p* < 0.001), and from 100 (96–104) beats minute^−1^ at T_0_ to 44 (34–57) beats minute^−1^ at T_44_ for DEX (*p* < 0.001). There was no relevant difference between treatments (*p* = 0.114). Respiratory rates and temperatures remained in the physiological range.

### 3.2. Phase 2 (T_45_–T_210_, ATI Versus SAL)

#### 3.2.1. Sedation

Sedation scores were lower in the dogs treated with ATI than SAL (*p* < 0.001). Time to sedation score ≤ 3 was 25 (15–25) minutes for ATI and 90 (68–118) minutes for SAL (*p* = 0.008, Figure 1b).

#### 3.2.2. Nociceptive Withdrawal Reflexes

The relative NWR thresholds recorded during the first phase (T_0_–T_44_) were not different for groups ATI and SAL (*p* = 0.293). The relative NWR thresholds were also similar at T_44_: 2.5 (1.9–3.3) for ATI and 2.3 (1.9–3.1) for SAL (*p* = 0.461, Figure 2a). Between T_45_ and T_210_, the relative NWR thresholds differed between groups (*p* < 0.001). Time to a relative NWR threshold ≤ 1.5 was five (3–6) minutes for ATI and 41 (28–97) minutes for SAL (*p* = 0.008). In dogs receiving ATI, a relative NWR threshold ≤ 1.5 was reached 18 (11–22) minutes before sedation scores returned to values ≤ 3 (*p* = 0.008, Figure 2b).

#### 3.2.3. Physiological Variables

After atipamezole administration, HR increased to 88 (80–107) beats minute^−1^ within 5 min. Respiratory rates and temperatures did not change.

### 3.3. Additional Observations on the Effect Offset

In the time interval T_45_–T_210_, sedation scores were higher for MED/SAL than for DEX/SAL (*p* = 0.02, Figure 3). Furthermore, time to sedation scores ≤ 3 were 155 (133–200) minutes for MED/SAL and 115 (88–135) minutes for and DEX/SAL (*p* = 0.057).

Relative NWR thresholds did not differ between MED/SAL and DEX/SAL subgroups (*p* = 0.241, Figure 4); similarly, time to relative NWR threshold ≤ 1.5 did not differ between the groups, at 116 (75–158) minutes for MED/SAL and 80 (73–118) minutes for DEX/SAL (*p* = 0.686). This difference of 30% was above the relevant endpoint of 20%, but at least 12 animals per group would have been required to confirm this result (Mann–Whitney test, unpaired, two-tailed, effect size = 1.3, α = 0.05, 1-β = 0.9, GPower 3.1, Germany).

In the SAL groups, the time to sedation scores ≤ 3 was 135 (113–163) minutes, whereas the time to relative NWR threshold ≤ 1.5 was 86 (73–142) minutes (*p* = 0.008), suggesting that sedation lasts longer than antinociception. A significant positive correlation between these two variables was found (*p* = 0.005, *p* = 0.87) and intercept-free linear regression had a slope of 0.81 (r^2^ = 0.74, Figure 5).

No relevant difference was observed between MED/ATI and DEX/ATI subgroups.

## 4. Discussion

### 4.1. Interpretation

The administration of a single IV dose of medetomidine (0.02 mg kg^−1^) or dexmedetomidine (0.01 mg kg^−1^) rapidly induced comparable sedation and antinociception in dogs. Dexmedetomidine did not induce significantly higher NWR thresholds than medetomidine. Both effects rapidly terminated after IM administration of atipamezole and the NWR thresholds decreased earlier than sedation scores.

In dogs, the sedative effects of dexmedetomidine administered alone [35] or as part of the racemic mixture [9,36] have already been reported. In our study, profound sedation (sedation scores ≥ 13 out of 21) and peak sedation scores were rapidly achieved after treatment administration. There was no difference in onset and quality of sedation between treatments. This is in contrast with Raszplewicz et al. [8], who reported that more dogs achieved profound sedation after medetomidine (0.01 mg kg^−1^) than after dexmedetomidine (0.005 mg kg^−1^) administered IM in combination with butorphanol. On the other side, in our study, the duration of drugs’ action was different: sedation lasted longer after medetomidine compared to dexmedetomidine. Similar results were reported in cats receiving high IM doses of alpha-2 agonists [37]. Other studies could not demonstrate any difference in sedation quality nor in duration of action between medetomidine and dexmedetomidine [3,4,15], but the sedation scores applied were poorly sensitive (numerical subjective scale from zero to three) compared to the scoring system used here [38]. Differences in duration of action probably depend on the doses administered, the route of administration and the co-administration of other drugs.

Differences in effect between dexmedetomidine administered alone and medetomidine as a racemic mixture are most probably due to the addition of levomedetomidine. When administered alone to dogs, levomedetomidine did not cause any behavioral effect [15] or decrease halothane minimum alveolar concentration (MAC) [39]. Only at high dosages, it moderately impaired the quality of the sedation induced by dexmedetomidine [15]. Therefore, the longer duration of sedation observed in our study after medetomidine is unlikely to be the result of a sedative effect from levomedetomidine. However, a pharmacokinetic interaction between the two enantiomers could occur, for instance, a slower elimination of dexmedetomidine in presence of levomedetomidine. While this has not been investigated yet, differences between dex- and levomedetomidine concentrations were observed after administration of the racemic mixture [13], suggesting an enantioselective pharmacokinetic.

Antinociceptive properties did not significantly differ between treatments in the present study. Previous results comparing analgesia induced by either medetomidine or dexmedetomidine were inconclusive. Longer lasting analgesia was reported with dexmedetomidine (0.02 mg kg^−1^ IV) compared to medetomidine (0.04 mg kg^−1^ IV) in response to toe pinch (assessed by a subjective score of the nocifensive response) [3]. Dexmedetomidine was not different from medetomidine at decreasing halothane MAC in dogs [39]. Likewise, no difference in nocifensive reaction to toe pinch was observed between dexmedomidine (0.04 mg kg^−1^ IM) and medetomidine (0.08 mg kg^−1^ IM) in both dogs [4] and cats [40]. These results would also benefit from being interpreted in perspective with pharmacokinetic studies. If sedation is hypothesized to be shorter when dexmedetomidine is administered alone due to lower plasma concentrations, as mentioned above, similar levels of antinociception could potentially support a stronger intrinsic analgesic efficacy for dexmedetomidine compared to medetomidine.

Anxiety may be difficult to observe in trained beagles. It could have potentially occurred during baseline measurements when the dogs were unmedicated, which could affect our results. Anxiety has been reported to increase central hyperexcitability and to potentially lower the NWR threshold [41,42], though this is controversial [43,44]. Efforts were made to acclimatize the dogs to the experimental room, staff and methodology. No sign of anxiety was noticed during the experiment. Baseline values for NWR threshold were comparable to previous studies [20] and to final values towards the end of the measurements. It is unlikely that anxiety affected the results presented here.

One objective of the present study was to apply a methodology that could potentially highlight differences in duration and efficacy between treatments. Using the automated continuous tracking device, even subtle variation in the NWR thresholds can be quantified over time. Moreover, the EMG-based NWR model is less likely to be influenced by sedation and muscle relaxation than visual scoring of a gross movement in response to stimulation. Finally, the evaluation can be repeated at high frequency (every 10 s) without risking habituation, sensitization or skin damage. Our results confirm that there is no difference in antinociceptive efficacy between medetomidine and dexmedetomidine in dogs at the doses used.

Another objective of this study was to compare the time courses of sedation and analgesia, in particular after atipamezole. A relevant observation was that the NWR thresholds returned to baseline before sedation, indicating that analgesic coverage is no longer guaranteed once sedation subsides. Intercept-free linear regression (r^2^ = 0.74, Figure 5) suggest that the duration of antinociception (relative NWR thresholds >1.5) approximates 80% of the duration of sedation (sedation score ≤ 3).

Antinociception was rapidly reversed by atipamezole at the dosage recommended by the manufacturer. The continuous NWR threshold determination tracked precisely the onset of antinociception reversal, which was approximately 6 min after atipamezole administration for both medetomidine and dexmedetomidine. This implies that appropriate analgesia needs to be considered whenever this antagonist is administered to animals experiencing pain.

### 4.2. Limitations

The comparison between MED and DEX followed a crossover design with eight subjects (paired) for the first 45 min only. Thereafter, the output variables (i.e., NWR threshold and sedation) in absence of reversal were compared between two groups of four dogs each (subgroups MED/SAL and DEX/SAL, no cross-over). The observations resulting from this comparison seem to indicate that DEX and MED may differ in duration of action more than in their potency, but this finding requires further validation.

Two female and six male intact dogs were used in the study. An equal sex distribution would have been preferable, as sex is known to potentially affect sensitivity to pain and thus nociceptive thresholds [45]. One female dog presented signs of proestrus during the second session of the trial. For this individual no difference was observed between the baseline values recorded during the two sessions but a certain influence on the nociceptive threshold course after drug administration cannot be excluded.

In the present study, antinociception was investigated using the non-invasive NWR model. A drug-related increase in the NWR threshold cannot be directly interpreted as clinical analgesia, as the complexity of clinical pain goes clearly beyond a spinal nociceptive reflex. Nevertheless, in humans the NWR threshold is known to correlate well to the threshold for pain perception [46], justifying the use of this model to assess pharmacologically induced analgesia in pain-free subjects in experimental settings.

### 4.3. Harm

No adverse events other than one female dog vomiting after both alpha-2-agonist injections was observed. Vomiting is common after administration of alpha-2 agonists as a result of their action at the chemoreceptor trigger zone [9,47,48].

## 5. Conclusions

Although sedation appeared to be slightly longer lasting after medetomidine, sedation quality and antinociceptive efficacy were similar for dexmedetomidine and medetomidine. Antinociception did not outlast sedation, and atipamezole was able to rapidly reverse both sedation and antinociception.

## Figures and Tables

**Figure 1 animals-10-01240-f001:**
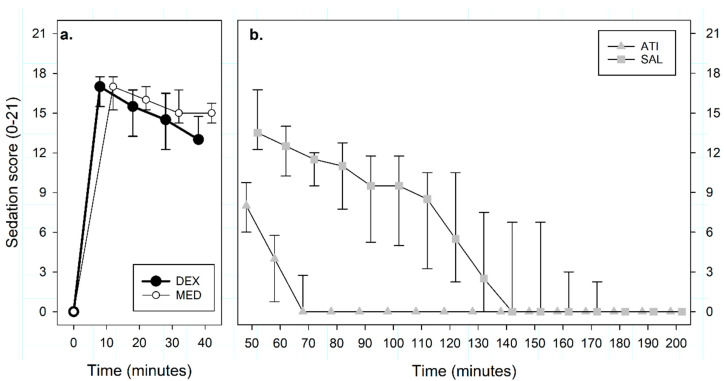
Median (interquartile range) sedation scores (from 0 to 21) recorded in 8 beagles after administration in a crossover design of (**a**) intravenous medetomidine (MED, 0.02 mg kg^−1^, n = 8) or dexmedetomidine (DEX, 0.01 mg kg^−1^, n = 8) at T_0_, and (**b**) intramuscular atipamezole (ATI, 0.1 mg kg^−1^) or an equivalent volume of saline (SAL) at T_45_. Data are presented slightly apart from the actual measurement time points to ease readability.

**Figure 2 animals-10-01240-f002:**
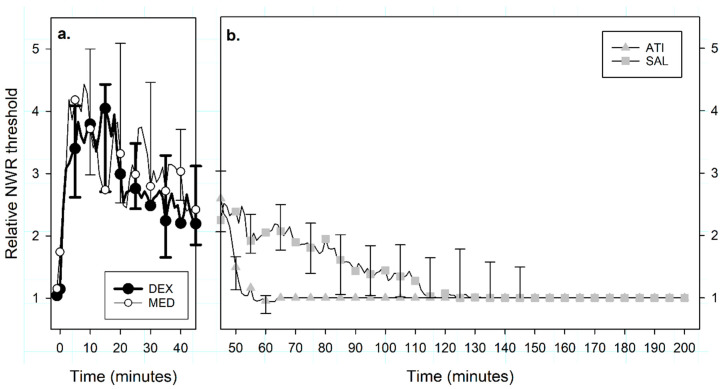
Median (interquartile range) relative nociceptive withdrawal reflex (NWR) thresholds recorded in 8 beagles after administration in a crossover design of (**a**) intravenous medetomidine (MED, 0.02 mg kg^−1^, n = 8) or dexmedetomidine (DEX, 0.01 mg kg^−1^, n = 8) at T_0_, and (**b**) intramuscular atipamezole (ATI, 0.1 mg kg^−1^) or an equivalent volume of saline (SAL) at T_45_. Data are presented only for representative time points (while actually recorded every minute) to ease readability.

**Figure 3 animals-10-01240-f003:**
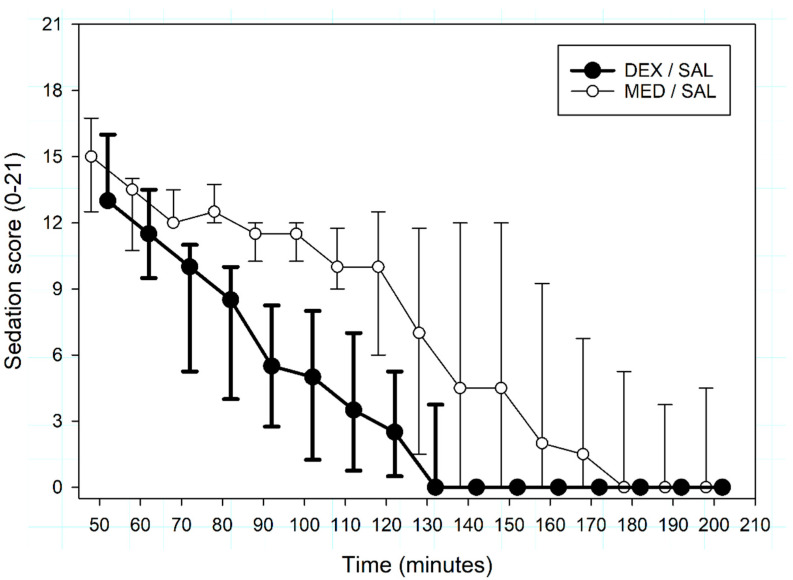
Median (interquartile range) sedation scores (from 0 to 21) recorded from T_45_ to T_210_ in 8 beagles sedated at T_0_ with either intravenous medetomidine (MED/SAL, 0.02 mg kg^−1^, n = 4) or dexmedetomidine (DEX/SAL, 0.01 mg kg^−1^, n = 4). Data are presented slightly apart from the actual measurement time points to ease readability.

**Figure 4 animals-10-01240-f004:**
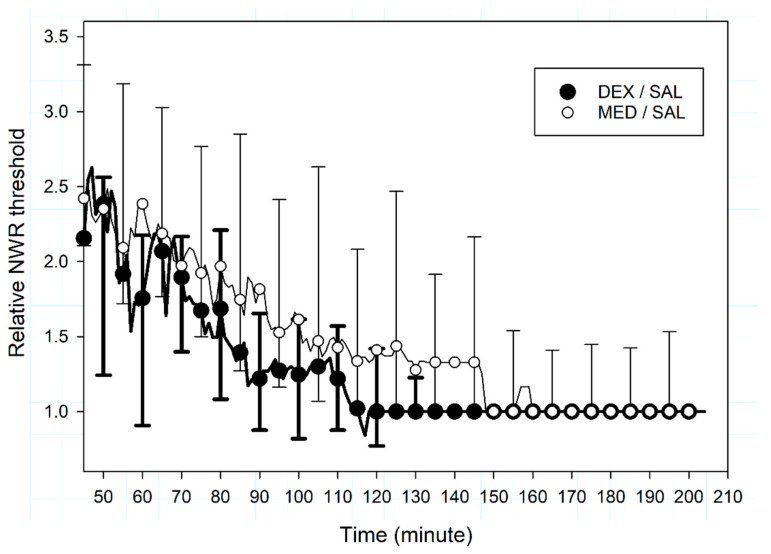
Median (interquartile range) relative nociceptive withdrawal reflex (NWR) thresholds recorded from T_45_ to T_210_ in 8 beagles after administration at T_0_ of either intravenous medetomidine (MED/SAL, 0.02 mg kg^−1^, n = 4) or dexmedetomidine (DEX / SAL, 0.01 mg kg^−1^, n = 4). Data are presented only for representative time points (while actually recorded every minute) to ease readability.

**Figure 5 animals-10-01240-f005:**
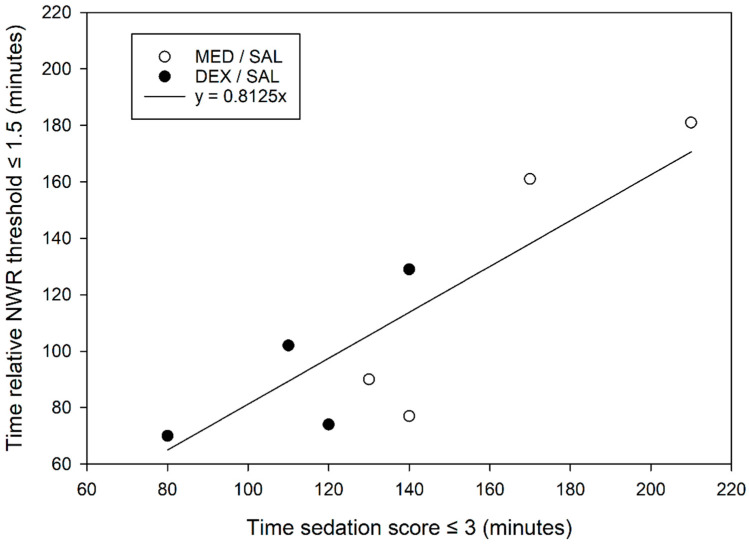
Intercept-free linear regression between the time to recover from sedation (sedation score ≤ 3) and the duration of antinociception (relative nociceptive withdrawal reflex (NWR) threshold ≤ 1.5) in 8 beagles after administration at T_0_ of either intravenous medetomidine (MED/SAL, 0.02 mg kg^−1^, n = 4) or dexmedetomidine (DEX/SAL, 0.01 mg kg^−1^, n = 4).

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
