# Peer review of "Effect of Medetomidine, Dexmedetomidine, and Their Reversal with Atipamezole on the Nociceptive Withdrawal Reflex in Beagles"

_animals, 2020, doi:10.3390/ani10071240_

Round 1
Reviewer 1 Report
Please find comments and suggestions for the authors in pdf file.

Author Response
We would like to thank the reviewer for the very helpful comments and the help at improving our manuscript. Please, find the changes highlighted in the revised manuscript, and consider that the revision has been performed also based on comments from other reviewers and from the editor. We hope that this revision of the manuscript has improved its quality.
Most minor comments have been implemented. Thank you very much.
Here are specific Point by Point answers.
R:" It should be made clear throughout the manuscript that Part 1 of the study is not novel but just a confirmation of already done work and shows that the NWR is also working for the two drugs studied."
A: We understand that equipotency for sedation and antinociception is not a new investigation and we believe the introduction and the discussion now clearly state previous research already existing. On the other hand, we explain that we expect the continuous NWR tracking to produce new data, which may or may not confirm conclusions from previous studies based on other methodology. Our aim was truly to compare the two drugs based on continuous electrical/EMG-NWR, which is an original new investigation.
R:" Please correct the figures: it is misleading to present part 1 and 2 in one graph".
A: We understand the comment and we have been trying to revise the figures accordingly but were finally not satisfied with the results when displaying both sedation and antinociception on the same graph. We believe that the figures presented as in the first submission are still less confusing (in particular for readability) presenting sedation score and NWR results separately even though we agree that this does not help at comparing durations of action.
R:" Also misleading in figures 1& 3 is that baseline should be at 45 min (as in Fig 4) – and the graph for part 2 starts at 50 min – please clarify. Are measurements taken at same time? Or is this to show SD? Please clarify."
A: Yes, a part of the legend has been erroneously removed and missing. The values displayed on the graph are slightly displaced for sedation, and partially omitted for NWR in order to ease readability otherwise the graphs were not readable due to the amount of information and the superposed IQR bars. Legends have been completed.
R:" Additional observations should be described in the discussion and not presented as results. Consider deleting parts of this paragraph, as n is low and these results were not amongst the predefined outcome variables."
A: Measured values are presented in the result section and their interpretation in the discussion part. We believe that these results, even though secondary to the predefined outcome variable, are of interest to be reported. Efforts have been made to inform the readership about their limitation and we believe that the readership is provided with the information to make their own interpretation of the value of these results.
R: "The main NOVEL finding of this study is that sedation outlasts analgesia after atipamezol administration. This should be emphasised in the first line of the discussion."
A: Changes have been made to support this main finding. Still, the article presents all the observed results from our methodology and the readership is able to make its own selection of the most relevant conclusions.
R:" Please consider to delete paragraph on anxiety in discussion (is not relevant for the outcome of this study)."
A: As mentioned in the discussion, occurrence of strong anxiety may be relevant for the outcome of the study (NWR threshold measurement) and we believe it is appropriate to briefly discuss this topic.
Reviewer 2 Report
Thank you for this interesting study comparing the anti-nociceptive and sedative effects of medetomidine and dexmedetomidine, and their reversal with atipamezole, in dogs. Use of the nociceptive withdrawal reflex model allowed for an elegant investigation of the anti-nociceptive effects of these drugs. My comments:
Line 22: Grammatical error “allow to”. Better wording would be “This model allows for quantification of…”
Line 54: Grammatical error “at half dose”. Better wording would be “at half of the administered dose of medetomidine…”
Line 122: Grammatical error “consisted in”. Should read “…consisted of..”
Lines 144-145: “An animal would have been excluded or replaced in case of abnormal physical examination of unexpected complication” – please revise grammar of this sentence.
Line 202: What is meant by “full blood”? Whole blood? But if stored in EDTA this suggests that plasma rather than whole blood was stored… Perhaps this should read “Blood samples (2mL) were obtained from the left IV catheter into EDTA collection tubes, at …”. Is it even necessary to provide this information given that the blood was not used to provide any data reported in this manuscript?
Line 225. The authors compared time for NWR to return to baseline and sedation score <4 between MED/SAL and DEX/SAL, however did they consider a similar comparison for MED/ATI vs DEX/ATI? Differences between these groups are unlikely, but it would be nice to also see this data
Line 230. As the authors already stated that the dogs “weighed” xxx; the words “body weight” at the end of the sentence are redundant / unnecessary.
Line 323: “without difference between treatments” – difference in what? This should be specified, or this part of the sentence removed/re-worded. Perhaps “…rapidly induced comparable sedation and antinociception in dogs.”
Author Response
We would like to thank the reviewer for the very helpful comments and the help at improving our manuscript. Please, find the changes highlighted in the revised manuscript, and consider that the revision has been performed also based on comments from other reviewers and from the editor. We hope that this revision of the manuscript has improved its quality.
Most comments have been implemented. Thank you very much.
R: "The authors compared time for NWR to return to baseline and sedation score <4 between MED/SAL and DEX/SAL, however did they consider a similar comparison for MED/ATI vs DEX/ATI? Differences between these groups are unlikely, but it would be nice to also see this data"
A: We agree with the reviewer. The values are very similar between groups and the low number of samples makes interpretation a little misleading. We added a sentence to report that there was no difference but decided not to provide all the numbers.
Reviewer 3 Report
Reviewer comments on Manuscript number: animals-854744
The present manuscript shows the effects of two alfa-2 adrenoreceptor-agonists (Medetomidine and dexmedetomidine), and their reversal with Atipamezole on the nociceptive withdrawal reflex and sedation in Beagle dogs
The manuscript is well written, and the information showed is interesting and relevant, as its main aim is to elucidate the common assumption of equipotency among medetomidine and dexmedetomidine.
Broad comments
The study is well designed, and the data showed is valuable, however, some parts of the manuscript need clarification before publication.
Weakness of the study: Age and weight of the animals. Use of experimental-trained dogs. Lack of Blood pressure measurements.
Specific Comments
Line 2-4. The title that best describes the study could be “Effect of medetomidine, dexmedetomidine, and their reversal with atipamezole on the nociceptive withdrawal reflex and sedation in Beagle dogs”
Lines 29-30 and 43-44. As these statements implies recommendations in clinical settings that were not part of this study, they should be removed from the simple summary and the abstract.
Lines 46-47. Keywords. Please add dexmedetomidine, instead antinociception if needed.
Line 113. Data of weight and age are missing at this point.
Line 114. The redaction of this phrase infers that perhaps 9 dogs instead 8 were housed.
Line 164: What was the plan for analgesic rescue, considering the possibility of discomfort after the reversion with atipamezole?
Line 177. Please describe exactly what signs of discomfort or pain were expected just before stopping the stimulation.
Line 187. Please describe what were the criteria to determine that the dogs became intolerant to the experiment.
Line 198. Considering the well -known hypertensive effects of both drugs, why did not you measure blood pressure?
Line 230. The dog’s weight and age should be reported in Materials and Methods. Did you considered that an effect of age may have contributed to bias some of your results? How many of the dogs were 8 years old? On the other hand, at least one of the dogs weighed 15.8 kg, that means overweight. Did you use the lean weight or total body weight for calculating the dose? In human beings, prolonged sedation may occur after dosing considering TBW instead lean body mass in obese patients.
Above statements should be considered in the Discussion section.
Thank you.
Author Response
We would like to thank the reviewer for the very helpful comments and the help at improving our manuscript. Please, find the changes highlighted in the revised manuscript, and consider that the revision has been performed also based on comments from other reviewers and from the editor. We hope that this revision of the manuscript has improved its quality.
Most comments have been implemented. Thank you very much.
R: "Considering the well -known hypertensive effects of both drugs, why did not you measure blood pressure?"
A: This was not the objective of our study. Arterial blood pressure was monitored occasionally with a non-invasive oscillometric device for the purpose of physical examination, but the methodology was not validated nor standardized. No result to report on this variable.
R:" The dog’s weight and age should be reported in Materials and Methods. Did you considered that an effect of age may have contributed to bias some of your results? How many of the dogs were 8 years old? On the other hand, at least one of the dogs weighed 15.8 kg, that means overweight. Did you use the lean weight or total body weight for calculating the dose? In human beings, prolonged sedation may occur after dosing considering TBW instead lean body mass in obese patients.
Above statements should be considered in the Discussion section."
A: As the dogs were recruited at the Moment of the study, Weight and ages are considered variable results of the inclusion rather than methodology parameters.
The age did not vary very much in our dogs (5 to 8, median 7) and major impact of this range of ages on NWR and sedation are not expected. Moreover, the crossover design allows to compare the treatments by pairs of same age (same individual receiving the two treatments).
Similar comment for the weight. It is right that particularly high or low BCS may influence PK and subsequent duration of effect. As none of the dogs was judged obese (BCS was always between 5 and 6 on the scale of 9, a sentence has been added in the method section), and the crossover would account for differences in weight, it is not expected that the differences of weight observed in the present study are a limitation for the interpretation of the result. We decided to avoid confusion in the discussion on this topic.